# Comparative Screening of the Structural and Thermomechanical Properties of FDM Filaments Comprising Thermoplastics Loaded with Cellulose, Carbon and Glass Fibers

**DOI:** 10.3390/ma13020422

**Published:** 2020-01-16

**Authors:** Alp Karakoç, Vibhore K. Rastogi, Tapani Isoaho, Blaise Tardy, Jouni Paltakari, Orlando J. Rojas

**Affiliations:** 1Department of Bioproducts and Biosystems, School of Chemical Engineering, Aalto University, FI-00076 Espoo, Finland; alp.karakoc@aalto.fi (A.K.); tapani.isoaho@gmail.com (T.I.); blaise.tardy@aalto.fi (B.T.); jouni.paltakari@aalto.fi (J.P.); orlando.rojas@aalto.fi (O.J.R.); 2Departments of Chemical & Biological Engineering, Chemistry and, Wood Science, 2360 East Mall, The University of British Columbia, Vancouver, BC V6T 1Z3, Canada

**Keywords:** additive manufacturing, fused deposition modeling, composite filaments, thermoplastic polymers, thermogravimetric analysis, differential scanning calorimetry

## Abstract

Additive manufacturing (AM) has been rapidly growing for a decade in both consumer and industrial products. Fused deposition modeling (FDM), one of the most widely used additive manufacturing methods, owes its popularity to cost effectiveness in material and equipment investment. Current efforts are aimed toward high load-bearing capacity at low material costs. However, the mechanical reliability of end-products derived from these compositions and their dependence on microstructural effects, have remained as major limitations. This is mainly owing to the unknown mechanics of the materials, including the reinforcing or filler components and their interphase/interface compatibility. For this reason, here we investigate the most relevant commercial polymeric materials used in composite filaments, associated phases and the characterization protocols that can guide component selection, screening and troubleshooting. We first present thermal analyses (thermogravimetric, TGA and differential scanning calorimetry, DSC) in relation to the constituent fractions and identify the type of polymer for uses in filaments production. The influence of various fillers is unveiled in terms of the crystallization behavior of derived 3D-printed parts. To understand the microstructural effects on the material strength, we carry out a series of tensile experiments on 3-D printed dog-bone shaped specimens following ISO standards. Simultaneously, real-time thermal energy dissipation and damage analyses are applied by using infrared measurements at fast frame rates (200 Hz) and high thermal resolution (50 mK). The failure regions of each specimen are examined via optical, scanning and transmission electron microscopies. The results are used to reveal new insights into the size, morphology and distribution of the constituents and interphases of polymer filaments for FDM. The present study represents advancement in the field of composite filament fabrication, with potential impact in the market of additive manufacturing.

## 1. Introduction

Advances in new and low-cost additive manufacturing (AM) combined with developments in polymer materials technologies related to AM play a critical role in the life cycle of today’s consumer and high-technology products [1,2,3,4,5]. Enjoying the high level of flexibility from the prototyping to the mass production phases, recent advances have been effective to replace conventional product development strategies. Meanwhile, such developments have allowed freeform fabrication and minimization of the reliance on expensive tools, providing more room for material recycling [6,7]. By means of these advances, it is possible to use thermoplastic polymers, which can be reused or recycled, as the matrix material binding the reinforcements in polymer composites. Compared to the given matrix in conventional thermosetting polymer composites, thermoplastic polymer composites are faster for fabrication and material toughening [8]. These characteristics make them attractive in mass-production industries with a focus on moderate operational conditions due to their low processing temperature and solvent resistance [9]. It is evident that the market share of the thermoplastic polymers and their composites has been rapidly increasing in correlation with the state-of-the art fabrication methods and processing strategies for their mass-production [10]. Additionally, for the low volume applications—e.g., in aeronautical and medical industries—thermoplastic polymers have been a frequent choice in the context of fused deposition modeling (FDM), which is a widely used AM method based on filament softening and extrusion through a nozzle [11,12]. The reason behind this choice is the wide range of commodity thermoplastics, which include polylactic acid (PLA), acrylonitrile butadiene styrene (ABS) and high-performance materials such as polyether ether ketone (PEEK) and polyetherimide (PEI), all of which can be effectively used with FDM [13,14]. However, for many of the FDM applications with thermoplastic composite filaments, the scientific foundations have been outrun due to process-centered industrial practices. In line with the extrusion and printing processes illustrated in Figure 1, the issues discussed so far can be addressed by controlling production process variables, including extrusion speed, heating and cooling cycles. During the formation and printing processes, crystallinity, mechanical and thermal characteristics of the filaments as well as constituents and compatibility at their interphases are affected to different extents [9].

In consideration to the extrusion and FDM processes, the present study investigates the mechanical and thermal characteristics of some of commercially available filaments, with matrix material composed of PLA, polyamide (PA6/66), or polyethylene terephthalate (PET) while using as reinforcement the so called “nanodiamond” (from uDiam-PLA or uDiam-PLA filaments), cellulose (UPM Formi-PLA), carbon and glass fibers (CF and GF, respectively). For a thorough characterization, thermogravimetric analysis (TGA) and differential scanning calorimetry (DSC) analyses were carried out. We identify the type of polymer used in their production processes, and determine the influence of various fillers on the crystallization behavior of their 3D printed parts. Thereafter, for understanding the microstructural effects on the material strength, a series of tensile test experiments were conducted on additively manufactured dog-bone specimens using simultaneous infrared (IR) thermal analysis. The failure regions of each specimen were finally examined with optical (bright field and polarized microscopy), and scanning electron microscopy (SEM) to gain a clear insight into the constituents, their interphases, and the propagated fracture post tensile testing. The results are expected to better connect industrial developments associated with thermosetting polymer composites and the scientific insights available in FDM applications.

## 2. Materials and Methods

### 2.1. Thermoplastic Polymer Composite Filaments for Additive Manufacturing

In the present study, four thermoplastic composite printing filaments (2.85 mm diameter) were used and abbreviated as UPM Formi-PLA, GF-PA6, CF-PET, Udiam-PLA. In addition, three single-component thermoplastic printing filaments, PLA, PA6/66 and PET, were also investigated. The detailed descriptions of the compositions and the proposed FDM printing conditions stated by the manufacturers are listed in Table 1 [15,16,17,18,19,20]. For the 3D printing of these filaments into dog-bone test specimens, an Ultimaker 3 Extended (Utrecht, Netherlands) printer was used with the recommended nozzle and printing bed temperatures. The selected 3D printer has a print bed moving only in the Z-axis while the print core (or extruder) can move in both X-and Y-axes, and is thus classified as Cartesian 3D printer [21]. The specimens were printed with 100% rectilinear infill in the XY-plane, also known as flat raster orientation, following the dimensional specifications, as shown in Figure 2 and provided in ISO 527-1:2012 plastics—determination of tensile properties [22,23]. Ultimaker Cura software (Version 4.3) was used for slicing and printing, through which the layer height was set to be 0.1 mm. Due to the abrasive nature of the reinforcing fibers and particles, the default nozzle system was replaced with the third-party 3D Solex Hardcore Pro print core and Everlast Ruby nozzle kit, where the nozzle size was 0.4 mm [24].

### 2.2. Thermogravimetric Analysis and Differential Scanning Calorimetry

The thermal properties of printing filaments (as received, without any further heat treatment or drying) were determined by thermogravimetric analysis (TGA Q500, TA Instruments, New Castle, DE, USA) and differential scanning calorimetry (DSC Q2000, TA Instruments, New Castle, DE, USA). TGA was used to determine the thermal degradation temperatures and to estimate the amount of filler material present in the composite filaments. A sample weight of 5–10 mg was heated from 30 to 600–800 °C at the rate of 10 °C/min in a flowing air atmosphere with a purge rate of 60 mL/min. Instead of performing the thermal degradation of printing filaments in a standard flowing nitrogen atmosphere, we intentionally carried out the degradation in air to simulate the 3D printing conditions that is often performed at ambient atmosphere.

DSC was performed on the printing filaments to (1) identify the type of polymer used in the production of filaments, based on the available literature, and (2) determine the influence of various fillers on the crystallization behavior of filaments and hence 3D printed parts. All experiments were performed under a constant nitrogen gas flow. The non-isothermal crystallization process was carried out according to the following procedure: Each sample weight of 5–6 mg (sealed in aluminum pans) was subjected to two heating cycles with an intermediate cooling. During the first heating cycle, the samples were heated from −40 °C to the given value depending on the suggested nozzle temperature, 180–300 °C (heating rate of 10 °C/min). After keeping them at 180–300 °C for 3 min, the given molten sample was cooled to −40 °C at a cooling rate of 10 °C/min and kept for another 3 min. Subsequently, the samples were heated again from −40 to 180–300 °C (heating rate of 10 °C/min). The first heating cycle was performed to erase the thermal history associated with the polymer processing and thermal properties such as glass transition temperature (*T_g_*), melting temperature (*T_m_*), melting enthalpies (Δ*H_m_*) and degree of crystallinity (*X_c_*), which were analyzed by using the second heating cycle [26,27,28]. However, in case of 3D printed polymers the first heating performed on the filaments is considered more appropriate and realistic for simulating the 3D printing conditions and hence determining the thermal properties of the 3D printed parts [29]. The melt crystallization temperatures (*T_mc_*) were determined from the cooling cycle. From the first heating cycle, the initial existing degree of crystallinity (*X_c_*_1*F*_) of filaments was evaluated by Equation (1), whereas, the evolution of overall crystallinity (*X_c_*_13*D*_) of 3D printed composites was calculated with Equation (2). The degree of crystallinity (*X_c_*_2*M*_) was also calculated from the second heating cycle (after removing the previous thermal history associated with the polymer processing) in order to determine the actual crystallinity of a polymeric material and calculated by Equation (3). The glass transition temperature (*T_g_*(ii)) of the filaments were also determined from the second heating cycle and compared with the glass transition temperature (*T_g_*(i)) obtained from the first heating cycle. TA Universal analysis software was used for the analysis of the TGA and DSC results.
(1)$$X_{c1F}=\;\frac{{\Delta H}_{m1}-{\Delta H}_{cc1}}{{\Delta H}_{m}^{o}\;\left(1-\frac{wt\%filler}{100}\;\right)}\times 100$$
(2)$$X_{c13D}=\;\frac{{\Delta H}_{m1}}{{\Delta H}_{m}^{o}\;\left(1-\frac{wt\%filler}{100}\;\right)}\times 100$$
(3)$$X_{c2M}=\;\frac{{\Delta H}_{m2}}{{\Delta H}_{m}^{o}\;\left(1-\frac{wt\%filler}{100}\;\right)}\times 100$$
where Δ*H_m_*_1_ and Δ*H_m_*_2_ represent the melting enthalpy of the crystals formed in the polymer during the first and second heating cycle, respectively, and Δ*H*_cc1_ is the enthalpy of cold crystallization formed during first heating. Δ*H°_m_* is the theoretical enthalpy value for a 100% crystalline polymer as follows: PLA = 93.7 J/g [30]; PA6 = 230 J/g [31]; PA6/66 = 240 J/g [32]; PET = 130 J/g [27] and wt% filler is the mass fraction of fillers present in the filament. *X_c_*_1*F*_ is the initial existing crystallinity of the printing filament and *X_c_*_13*D*_ is considered as the overall crystallinity of 3D printed part evolved after 3D printing (from first heating cycle). *X_c_*_2*M*_ is the actual crystallinity of a polymeric material calculated from the second heating cycle.

### 2.3. Mechanical and Thermal Characterization Procedure

Mechanical testing procedure was based on the tensile strength of dog-bone shaped specimens, which followed the geometry and test method specifications stated in ISO 527-1:2012 plastics—determination of tensile properties [23]. Tensile tests were carried out at room temperature (20 °C), at a relative humidity of 60%. The test speed was 5 mm/min with nominal initial grip to grip separation of 110 mm for each specimen [33]. The force transducer, with a limit of 20 kN, was used with a mechanical gripping system on a Zwick tensile testing frame. In order to measure the real-time thermal energy dissipation and to identify the thermomechanical characteristics of the damage, FLIR A655SC infrared (IR) measurement system was also used with the frame rate of 200 Hz, optical resolution of 640 × 480 px^2^, and thermal resolution of 50 mK.

### 2.4. Optical Measurement Systems

Optical microscopy was used to gain a deeper insight into the filament composition, for which the images were obtained on a Nikon microscope (Tokyo, Japan) in transmission and reflection mode with or without cross-polarizers. A JVC KY-F55BE camera with a resolution of 752 × 582 was used. For the observation and assessment of the fractures after tensile testing, a field emission scanning electron microscope (FE-SEM; Zeiss SigmaVP, Jena, Germany) operating at 1.6 kV and a working distance of 4 mm was used. Prior to imaging the samples were sputtered with a 4 nm layer of platinum-palladium alloy.

## 3. Results

### 3.1. Thermoplastic Polymer Composite Filaments for Additive Manufacturing

The thermal degradation of the different printing filaments at temperatures between 23 and 600–800 °C is illustrated by the TGA data (air atmosphere) included in Figure 3. According to the weight loss profiles (Figure 3a), the thermal degradation of the printing filaments occurs in two steps, except for CF-PET, which degraded following three steps. The thermal degradation temperature at each step was determined from the derivative thermogravimetric analysis (Figure 3b), where *T_d_*_1_, *T_d_*_2_ and *T_d_*_3_ represents the degradation temperature at first, second and third step (Table 2). Comparing neat and composite filaments of PA6/66 and PET, it is evident that PLA and its composites are less thermally stable. As such, the initial degradation temperature (*IDT*) was determined for each filament and compared to determine the onset of thermal degradation (Table 2). Clearly, PA6/66 and GF-PA6 are the most thermally stable materials (*IDT* of 390.34 and 393.34, respectively), followed by PET and CF-PET, which displayed an intermediate stability (*IDT* of 381.47 and 386.08, respectively). The results compared to those of PLA, UPM Formi-PLA and Udiam-PLA with lowest stability due to an early onset of degradation (*IDT* of 321.11, 310.61 and 334.85, respectively). Among all composite filaments, only PLA containing cellulose fibers (UPM Formi-PLA) degraded slightly earlier than pure PLA, which is attributed to the early degradation of hemicellulose in the temperature range of 220–280 °C [34]. In contrast, other composites containing nanodiamond particles, glass or carbon fibers performed similarly or slightly better than their reference polymers, potentially due to better thermal stability of the fillers. In addition, the presence of any residual volatile solvent, bound and unbound water in the printing filaments, was also determined from TGA by measuring the weight loss at 220 °C, *WL*_220 °C_ (Figure 3a, inset). As such, PLA, PET and CF-PET resulted in minimum weight loss at 220 °C (<0.19%) inferring the absence of any solvents. However, UPM Formi-PLA, Udiam-PLA, PA6/66 and GF-PA6 resulted in higher weight loss (1.10%, 0.87%, 1.62% and 1.07%, respectively) that indicates the presence of solvents. For UPM Formi-PLA, the weight loss should be attributed to the adsorbed water on the hygroscopic cellulose fibers from the ambient environment. Finally, from the weight loss curves (Figure 3a) the amount of filler content was determined at the first, second and third step (*WL*_1_, *WL*_2_ and *WL*_3_) and the residual ash (Table 2). As such, a filler content similar to the data provided by the respective producer was obtained: UPM Formi-PLA contains 14.70 wt.% cellulose fibers (15–50 wt.% stated by the producer), Udiam-PLA contains 14.62 wt.% nanodiamond particles (produced data not available), GF-PA6 contains 29.5 wt.% glass fibers (30 wt.% as stated by the producer) and CF-PET contains 15.74 wt.% carbon fibers (15 wt.% according to the producer).

The crystallization behavior of neat and composite printing filaments were investigated by DSC non-isothermal crystallization using the first and second heating and cooling thermographs, as shown in Figure 4. Table 3 summarizes the thermal parameters from the DSC curves, such as glass transition temperature (*T_g_*_(i)_, first heating and *T_g_*_(ii)_, second heating), cold crystallization temperature (*T_cc_*), cold crystallization enthalpy (Δ*H_cc_*), melting enthalpy (Δ*H_m_*), onset crystallization temperature (*T_onset_*), melt crystallization temperature (*T_mc_*) and derived parameters such as half crystallization time (t_1/2_), crystallization rate (R), crystallinity of filament (*X_c_*_1*F*_), crystallinity of 3D product (*X_c_*_13*D*_) and actual crystallinity of polymeric material (*X_c_*_2*M*_). The half crystallization time and crystallization rate were obtained following the literature [35] where
(4)$$t_{1/2}=\frac{T_{onset}-T_{mc}}{\varphi }.$$
(5)$$R=\frac{1}{t_{1/2}}.$$

Here, *T_onset_* is the temperature at the onset of crystallization during the cooling cycle, *T_mc_* is the peak temperature of the crystallization endotherm, *φ* is the cooling rate at 10 °C/min, *t*_1/2_ represents the time when 50% of the polymer already being crystallized and *R* represents the crystallization rate.

From the first heating cycle (Figure 4a), similar glass transition temperatures, *T_g_*_(i)_ were observed for PLA, UPM Formi-PLA and Udiam-PLA, indicating that no plasticizer was used in the manufacture of the filaments, which would have otherwise shifted the *T_g_*_(i)_ towards lower values. From Table 3, similar glass transition *T_g_*_(ii)_ temperatures were observed when determined from the second heating cycle, inferring that the slight moisture uptake by the cellulose-containing systems and those with nanodiamonds (as determined previously from the weight loss at 220 °C, TGA) did not affect the bulk properties of PLA. However, addition of fillers significantly affected the crystallization of PLA that occurred initially at 114 °C (*T_cc_*) for neat PLA and latter shifted to lower temperatures of 91.2 and 79.8 °C upon addition of cellulose fibers and nanodiamond particles (UPM Formi-PLA and Udiam-PLA, respectively). Clearly, the crystallization of PLA composites becomes easier in the presence of fillers, which act as nucleating agents. Notably, from the cooling cycle (Figure 4b), a well-developed melt crystallization peak was observed for Udiam-PLA, around 120 °C, which compares with an under-developed peak at 30 °C for UPM Formi-PLA. In addition, no cold crystallization peak (*T_cc_*) was observed for Udiam-PLA during the second heating cycle (Figure 4c), indicating the complete crystallization of PLA during the previous cooling cycle. In contrast, a fully developed cold crystallization peak was observed for UPM Formi-PLA, indicating partial crystallization of PLA, in response to the coarser and heterogeneous morphology of cellulose fibers that hindered PLA crystallization, due to restriction in PLA polymeric chain movement. Therefore, nanodiamond particles, which promoted an early crystallization and hence improved PLA crystallization behavior, are considered a better and more efficient nucleating agent than cellulose fibers. Interestingly, the melting temperature (*T_m_*) of Udiam-PLA was around 170 °C while for PLA and UPM Formi-PLA it was around 150 °C. Similar observations have been reported for two different grades of PLA differing in D-isomer content, where, grade with 1.5% D-isomer results in melting temperature of 170 °C and grade with 4% D-isomer results in a melting temperature of 150 °C [28]. Therefore, a similar grade of PLA with around 4% D-isomer content was used to produce neat PLA and UPM Formi-PLA filaments (might favored for 3D printing due to low melting temperature), while a PLA grade with 1.5% D-isomer content was used in Udiam-PLA filament. For UPM Formi-PLA, the melting peak splits into a lower temperature peak (*T_m_*_1_) that refers to the melting of the disordered crystal form, while the second peak at higher temperature (*T_m_*_2_) corresponds to the melting of more ordered forms of crystals [28], whereas, no splitting was observed for Udiam-PLA. The splitting may lead to ambiguities in the crystalline structure of 3D printed product and may reduce their mechanical properties. Surprisingly, a third melting peak (*T_m_*_3_) was also detected for Udiam-PLA, around 220 °C, that might be due to the formation of more stable crystals of PLA in the presence of nanodiamond particles, which act as effective nucleating agent. This might be a reason for having slightly higher *IDT* compared to neat PLA (Table 2).

For PA6/66 and GF-PA6, no cold crystallization peak (*T_cc_*) was observed from the heating cycle (Figure 4a), inferring that no new crystals were formed during the heating as the crystallization already reached its maximum after filament processing (during cooling). For PA6/66, two melting peaks (*T_m_*_1_ and *T_m_*_2_) were observed comparing the single peak for GF-PA6 and was attributed to the presence of PA6 and PA66 polymer with different melting points. A copolymer of PA6 with a fraction of PA66 is favored for 3D printing application, as it leads to a decrease in melting point and crystallization, enhances clarity and comprehensive mechanical properties, it also reduces the average material cost [32]. The copolymer filament of PA6/66 used in this study was blended in a ratio of 80:20, as determined from the very similar thermal properties (*T_m_*_1_, *T_m_*_2_ and *T_mc_*) of PA6/66 mentioned in the literature [32]. In contrast to PLA and its composites, a significant difference was observed in the glass transition temperature of PA6/66 and GF-PA6 from the first and second heating cycle (*T_g_*_(i)_ and *T_g_*_(ii)_, Table 3). As such, the lower value of *T_g_*_(i)_ compared to *T_g_*_(ii)_ indicates the presence of a residual compound (water/solvent; as determined previously from the weight loss at 220 °C, TGA) acting as a plasticizer for the polymer matrix. Such solvent evaporates after the first heating cycle and latter resulted in higher *T_g_*_(ii)_. For GF-PA6 composite filament, similar thermal properties (second heating, *T_g_*
_(ii)_ = 67.1, Tm = 206 and Tmc = 171 °C) were observed comparing the neat PA6 (second heating, *T_g_*
_(ii)_ = 65.9, *T_m_* = 211 and *T_mc_* = 170 °C) as mentioned in the literature [36]. Due to the similar *T_mc_* obtained after adding glass fibers in the PA6 matrix, they do not act as nucleating agent for PA6 matrix, which otherwise resulted in composites with higher *T_mc_*. Thus, glass fibers only produced a reinforcing effect in the PA6 matrix.

For PET filaments, the glass transition temperature is clearly observed, with no signs of melting and melt crystallization peak (Figure 4a,b), which suggests amorphous PET. For 3D printing applications, PET is often copolymerized by replacing the ethylene glycol groups of PET with the 1,4-cyclohexanedimethanol (CHDM), leading to poly(ethylene glycol-co-1,4-cyclohexanedimethanol terephthalate) (PETG), a modified PET with reduced melting temperature and crystallinity [27]. From the literature, the PETG used in this study consists of 30 wt.% CHDM, which removes the potential for PET chains to form ordered structures [27], and therefore results in a completely amorphous rather than semi-crystalline polymer. For the CF-PET composite, a semi-crystalline PET was used without any blending (100/0), as determined by the similar thermal properties (second heating, *T_g_*_(ii)_ = 74.84, *T_m_* = 245.27 °C) comparable to the value found in the literature (second heating, *T_g_*_(ii)_ = 76.30, *T_m_* = 242.50) [27]. However, higher *T_mc_* (210.55°C) was obtained for CF-PET comparing the neat PET (*T_mc_* = 185.40 °C), [27] which definitely point towards the nucleating effect of carbon fibers in enhancing the crystallization of PET. In addition, no cold crystallization peak (*T_cc_*) was observed for CF-PET during the second heating cycle (Figure 4c) indicating the complete crystallization of PET during the previous cooling cycle; hence, the carbon fibers proved to be efficient nucleating agents.

From Table 3, three types of crystallinities can be discussed, e.g., the existing crystallinity of printing filaments (*X_c_*_1*F*_), overall crystallinity of 3D printed part (*X_c_*_13*D*_) and actual crystallinity of the polymeric filament material (*X_c_*_2M_). For 3D printing applications, filaments are often quenched before spooling to partially restrict the crystallization of the polymer or, in other words, to maintain the polymer in the amorphous state where it is easily soften/melt during the 3D printing. Therefore, for most of the filaments (such as PLA, UPM Formi-PLA, Udiam-PLA and CF-PET) a lower degree of crystallinity (*X_c_*_1*F*_) is expected, which fully evolves only during the 3D printing by the action of heat and results in the formation of new crystals in the 3D printed part (*X_c_*_13*D*_). However, there are some polymer filaments like PA6/66 and GF-PA6 where quenching did not work or the polymer was annealed to achieve the required properties. As such, no new crystals were developed during the 3D printing (Figure 4a), i.e., full crystallization was already achieved during the filament processing (cooling) that results in *X_c_*_1*F*_ equal to *X_c_*_13*D*_. The actual crystallinity (*X_c_*_2M_) of a polymeric material (printing filament) is often determined from the second DSC heating cycle [26,27,28], after removing the previous thermal history during the first cycle. Consequently, differences were observed in the values of *X_c_*_13*D*_ and *X_c_*_2M_, where, the crystallinity of 3D printed part (*X_c_*_13*D*_) was usually slightly higher or equal to the actual crystallinity of polymeric material (*X_c_*_2M_), depending on the previous thermal history (Table 3).

The effect of fillers on the actual crystallinity of polymeric materials (*X_c_*_2M_) can be determined by comparing it with the crystallinity of their respective matrix polymers. As such, addition of cellulose fibers in PLA (UPM Formi-PLA) resulted in a reduced *X_c_*_2M_ (19.7%) if compared to the *X_c_*_2M_ (21.3%) of neat PLA (see Table 3). This might be due to an overloading of cellulose fibers (15 wt.% from TGA) added as filler in the PLA matrix, which hindered the flexibility of PLA polymeric chains and led to a limited crystallization of PLA. Similarly, for GF-PA6 and CF-PET composites with glass fibers (30 wt.%) and carbon fibers (15 wt.%), a reduction in *X_c_*_2M_ (24.8% and 32.2%, respectively) was observed compared to the *X_c_*_2M_ of the matrices, PA6 and PET (28% [36] and 35% [27], respectively). From previous observations, glass fibers are more likely to act only as a reinforcing agent for PA6 while only carbon fibers are considered as a nucleating agent for PET. A higher loading of fillers in the polymer matrix, however, results in reduction of *X_c_*_2M_ due to restricted polymer chain movement. In contrast, for Udiam-PLA, a significant improvement in *X_c_*_2M_ (52%) is observed compared to the *X_c_*_2M_ (36%) of neat PLA [30]. It is interesting to note that both cellulose fibers and nanodiamond particles were loaded at 15 wt.% in the PLA matrix but only nanodiamond particles with fine and homogeneous morphology acted as effective nucleating agent for PLA when compared to cellulose fibers, with coarser and heterogeneous morphology. The detailed explanation of the effect of filler size on the crystallinity of the composite could also be investigated by the activation energy required by fillers to crystallize the polymer, a subject for further studies.

For 3D printing applications, both GF-PA6 and CF-PET are considered an ideal choice as printing filaments, displaying the lowest half crystallization time, *t*_1/2_ and highest crystallization rates, *R* (Table 3). However, GF-PA6 may be preferred over CF-PET due to lower melting temperature that is often associated with better energy efficiency in 3D printing.

### 3.2. Tensile Tests and IR Thermography

The tensile test specimens were printed in batches of three to provide the same environmental and printing conditions, thus eliminating any effect of unknown manufacturing parameters. As listed in Table 4, most of the tested specimens exhibited significantly lower Young’s moduli (37–79% relative to the values provided by the manufacturers [15,16,17,18,19,20]). The main reason for the difference related to the procedure used for testing the specimens. For convenience, some of the manufacturers produce both pellets and filaments of same compositions and provide the characteristics by using the injection molded test specimens, which is the only way to present the pellet characteristics. It is known that the injection molding uses external forces to reduce the voids and defects resulting in final products that exhibit better performance than the ones manufactured with FDM [13,37]. As observed in the SEM images detailed in the following section, raster orientations for the fibers, which were not aligned under loading during the tensile testing, was another reason for the reduced value of the mechanical characteristics of the specimens manufactured with FDM [38]. However, exceptions were the experimental results obtained from PA6/66, which exceeded the provided Young’s modulus by 120% [15]. The ultimate tensile strengths (UTS) of filaments comprising the polymeric matrices (PLA, PET and PA6/66) were higher than those quoted by manufacturers while the UTS for GF-PA6 composite filaments was similar to the manufacturer’s figures. For the rest of the investigated materials, the measurements indicated lower values than quoted.

During the tensile testing measurements, IR thermography was simultaneously carried out to better understand the thermomechanical characteristics of the materials. To eliminate any disturbance in the acquisition of the IR thermographs, e.g., from the surroundings and also to provide a reasonable contrast, the background during the measurement was covered with a wetted copier paper of 80 g/m^2^. Based on Δ*T*_Break_ (listed in Table 4), which refers to the difference of temperatures at the instant of failure (break at ultimate tensile strength UTS) and the initial configuration, and temperature profiles of Figure 5, filaments of the polymeric matrix phase (PLA, PET and PA6/66) are shown to dissipate high thermal energy in a correlation with the measured strain values *ε*_UTS_ and *ε*_Break_ in the plastic region. This effect, especially for PA6/66, was evident in the thermal images shown in Figure 6. In case of the composite filaments, GF-PA6 and uDiam-PLA, they dissipated quite high thermal energies although they display brittle behavior (see the close *ε*_UTS_ and *ε*_Break_ strain values). Especially for UPM Formi-PLA, Δ*T*_Break_ values were measured to be the lowest among the others, which can be related to low bonding characteristics between the fibers and the polymeric interface and, more importantly, the presence of fibers aggregates (please, see the SEM images in the following section for details).

### 3.3. Optical Measurements

Microscopy was used to for morphological elucidation of the components dispersed in the filaments as well as to gain mechanistic insights into their strain-dependent behavior. Optical microscopy in reflection and transmission mode revealed large-scaled grain for the UPM Formi-PLA filaments (Figure 7a, ca. 200 µm × 50 µm). The brown color observed suggest substantial amounts of residual lignin. The grains were brighter (white) when observed between cross polarizers (Figure 7b) highlighting their birefringent nature. The latter is generally associated with cellulose’s crystalline domains. Such an effect was not observed for filaments containing glass or carbon fiber (Figure 7c,d).

Observations by scanning electron microscopy (SEM) of the fractures obtained after tensile tests revealed information on the composites as well as their behavior under strain. For instance, a high volume fraction of fiber in the UPM Formi-PLA filaments was observed (Figure 8e). The distinction between the fibers and the matrix was rather difficult and there was no clear evidence of debonding between the fibers and the polymeric interface (Figure 9f). Large fibrous aggregates were present, as also observed by bright field microscopy between cross-polarizers. The fracture was multi-scaled with features from the submicrometer scale to >200 µm. This is in contrast with the matrix polymer, where the fracture was clean and smooth (Figure 8a). Multi-scale fracture propagation was also observed for CF-PET, and GF-PA6 composites (Figure 8c,d), respectively. The Udiam-PLA composite filaments displayed a rather smooth fracture (Figure 8b). The largest fracture features were observed for UPM Formi-PLA followed by GF-PA6 and CF-PET.

Observing the interface at the single reinforcing component scale, the UPM Formi-PLA fibers appeared well bound to the polymeric matrix (Figure 9f). A good polymer-additive interface was observed for Udiam-PLA (Figure 9a,b), where the fillers appeared as flakes with a nanometer-scaled thickness and highly polydisperse planar dimensions, ranging from 1 to 10 µm.

Interestingly, bright spots could be observed when imaging uDiam-PLA between cross polarizers, suggesting filler’s crystalline attributes (Figure 10). The carbon fiber interface with the polymer was also good in the case of filaments containing carbon fibers (Figure 9c,d). This strong interaction between the CF and PET must be the reason for CF to act as an effective nucleating agent for PET matrix. The CF cross sectional diameter appeared rather homogeneous, ranging from 4 to 10 µm. The polymer can be seen clearly on the fractured carbon fiber, up to the actual cross sections of the fractured filler. Nevertheless, some debonding was observed. In contrast, the glass fiber had a poor interaction with the PA6 in the GF-PA6 composites (Figure 9e), with the glass fibers largely protruding from the fractured area and clearly showing debonding around the polymer-GF interphase. This observation was in line with the previous DSC measurements where no nucleating activity was associated with the glass fibers, which only acted as reinforcing agent in the PA6 matrix. The diameter of the glass fibers (5 µm) was observed to be rather homogeneous.

## 4. Conclusions

In the present study, some of the commercially available filaments comprising neat polymers and respective composites were investigated for their thermo-mechanical properties. From thermogravimetric analysis (TGA), PA6/66 and GF-PA6 were found to be the most thermally stable materials. PET and CF-PET showed an intermediate stability while PLA, UPM Formi-PLA and Udiam-PLA displayed the lowest stability. However, PA6/66, GF-PA6, UPM Formi-PLA and Udiam-PLA are the most water sensitive materials if compared to PLA, PET and CF-PET. For composite filaments, a similar filler loading was determined as that quoted by the manufacturer, which was further confirmed by TGA. From differential scanning calorimetry (DSC), the polymeric materials of all printing filaments were identified after comparing their thermal properties (*T_g_* and *T_m_*). As such, two different PLA grades were identified based on different melting temperatures (170 and 150 °C, D-isomer content of 1.5% and 4%, respectively), a copolymer of PA6/66 identified with 80/20 blending ratio, and PET with a completely amorphous nature (known as PETG) obtained after blending 30 wt.% CHDM. The results confirm the practice of blending, modifying tor quenching the standard polymers to reduce their melting temperature and to achieve better energy efficiency. It is demonstrated that nanodiamond particles and carbon fiber (CF) acted as an efficient nucleating agent (higher *T_mc_*). In contrast, cellulosic and glass fibers (GF) were not effective in enhancing the crystallization of the matrix. Notably, only nanodiamond particles with fine spherical geometry resulted in an increased the crystallinity of polymer matrix, while, cellulose, carbon or glass fibers, with large fibrous morphologies, resulted in a reduction of composite crystallinity. This is owing to the more restricted polymeric chain movement. Microstructural effects were related to the failure regions of the tested specimens. A good polymer-additive interface was observed for all fillers apart from glass fibers, which displayed no clear attachment to the matrix. UPM-Formi was revealed to contain cellulosic fibers while Udiam contained crystalline, micro-scaled flakes with a nanometer thickness. All fillers increased tortuosity of the fracture propagation across the filaments, aside from Udiam, potentially due to its smaller dimensions. Overall, the present study shows that there is a great deviation between the experimental strength data and the values quoted by the manufacturers, which is presumably related to the printing conditions, i.e., printing temperature, speed, ambient conditions, infill pattern and percentage [39,40]. Another factor is related to variation in test specimens produced with injection molding. Since the voids are reduced with pressure and temperature, the strength properties of the specimens are definitely higher than the ones obtained from FDM. The considerations brought forward in this study are expected to result in further progress, both in reverse engineering to associate a deeper science with ongoing industrial efforts and in the formation of high-performance filaments for FDM.

## Figures and Tables

**Figure 1 materials-13-00422-f001:**
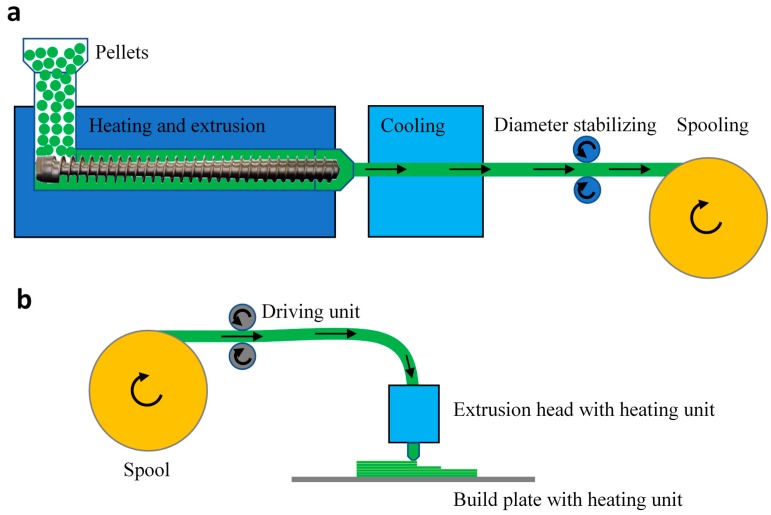
Schematic illustration of the cycle from pellets to printed product: (**a**) 3-D printing filament extrusion process and (**b**) fused deposition modeling (FDM).

**Figure 2 materials-13-00422-f002:**
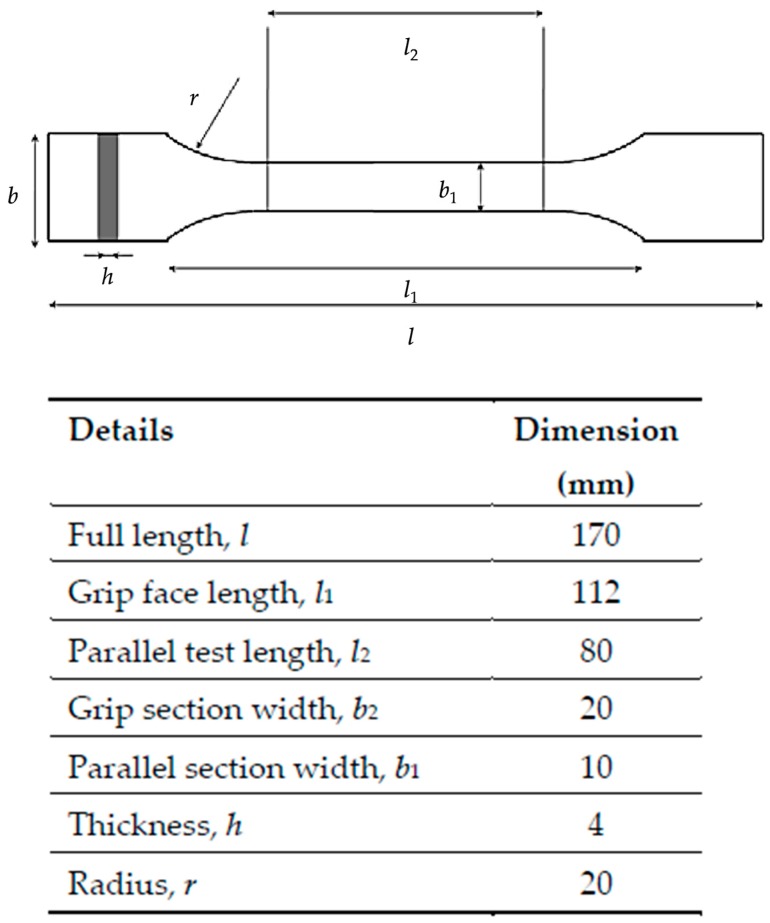
Dimensional specifications of the specimen according to ISO 527-1:2012 [23,25].

**Figure 3 materials-13-00422-f003:**
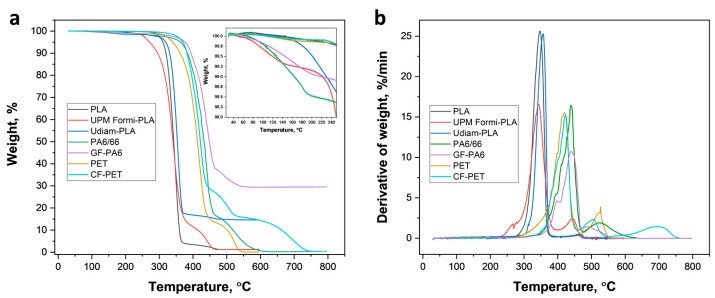
Thermogravimetric analysis representing (**a**) weight loss, inset (details up to 250 °C) and (**b**) differential weight loss of different printing filaments.

**Figure 4 materials-13-00422-f004:**
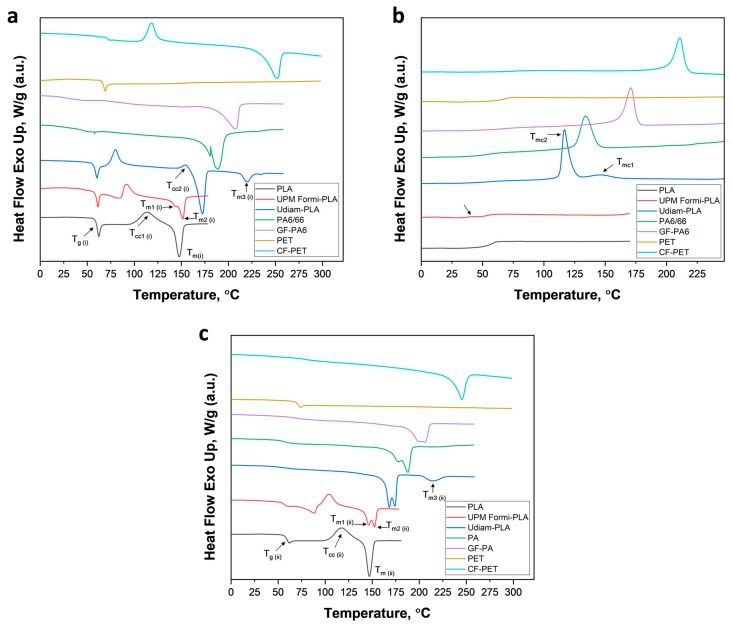
Non-isothermal differential scanning calorimetry (DSC) curves for different printing filaments showing (**a**) first heating cycle, (**b**) cooling cycle and (**c**) second heating cycle.

**Figure 5 materials-13-00422-f005:**
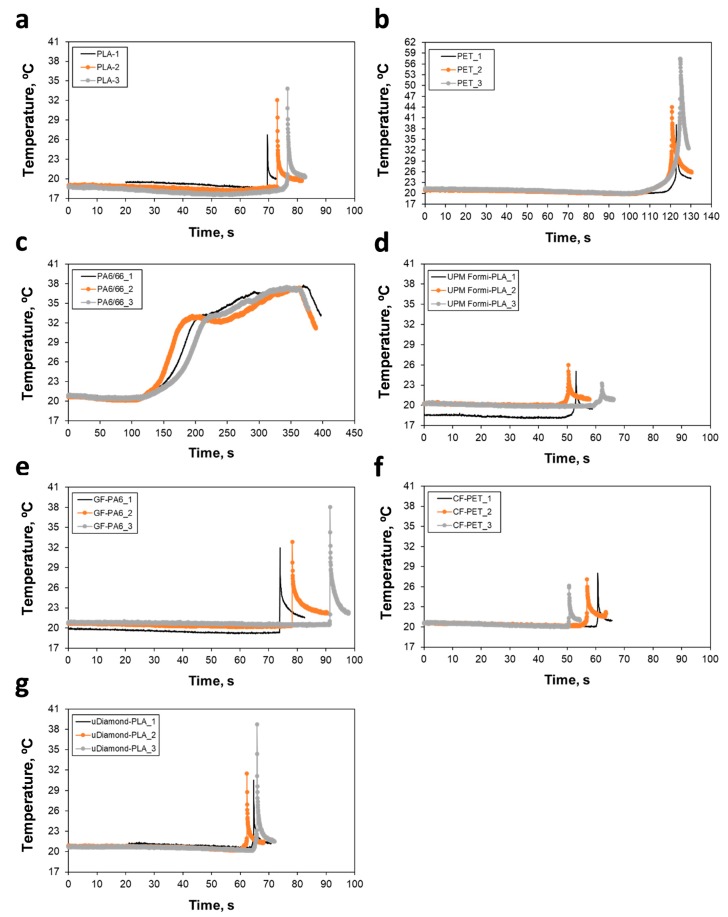
Temperature profiles measured during the tensile testing with IR measurement system: (**a**) PLA, (**b**) PET, (**c**) PA6/66, (**d**) UPM Formi-PLA, (**e**) GF-PA6, (**f**) CF-PET and (**g**) uDiam-PLA.

**Figure 6 materials-13-00422-f006:**
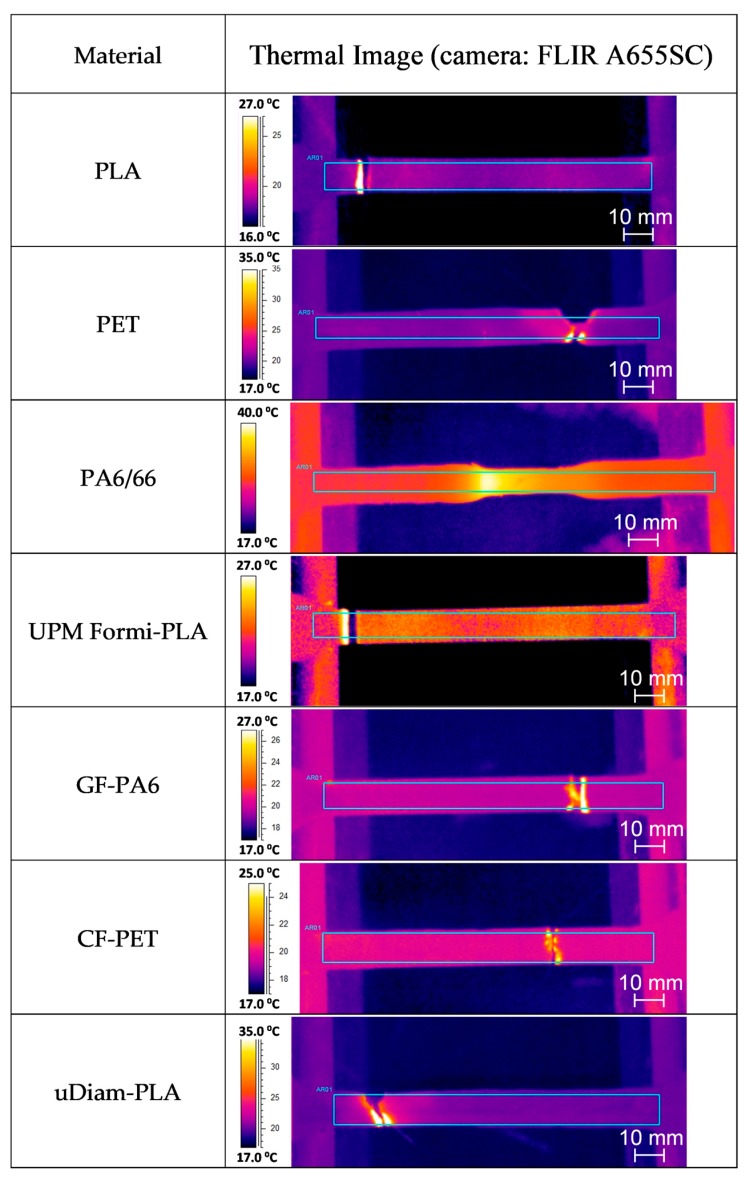
Infrared thermography for some of the investigated specimens.

**Figure 7 materials-13-00422-f007:**
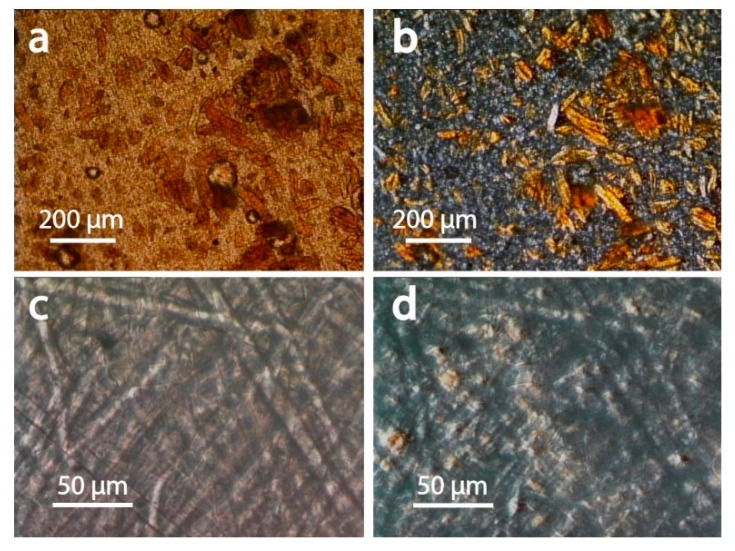
Transmission microscopy images obtained for (**a**) UPM Formi-PLA based composites and (**c**) GF-PA6 based composites. In (**b**,**d**) the respective images are obtained with the sample between cross polarizers highlighting the crystalline nature of the fibers in UPM Formi-PLA in (**b**) and of the amorphous nature of the glass fibers in (**d**).

**Figure 8 materials-13-00422-f008:**
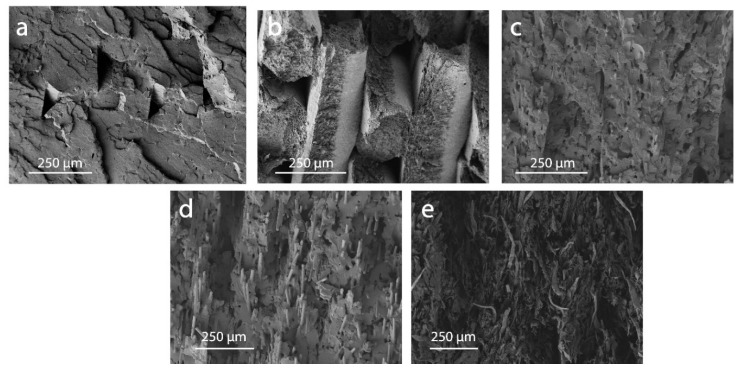
Example of fractured surfaces as obtained after tensile tests observed by SEM for (**a**) neat polymeric matrices and the respective composites, (**b**) Udiam-PLA, (**c**) CF-PET, (**d**) GF-PA6 and (**e**) UPM Formi-PLA.

**Figure 9 materials-13-00422-f009:**
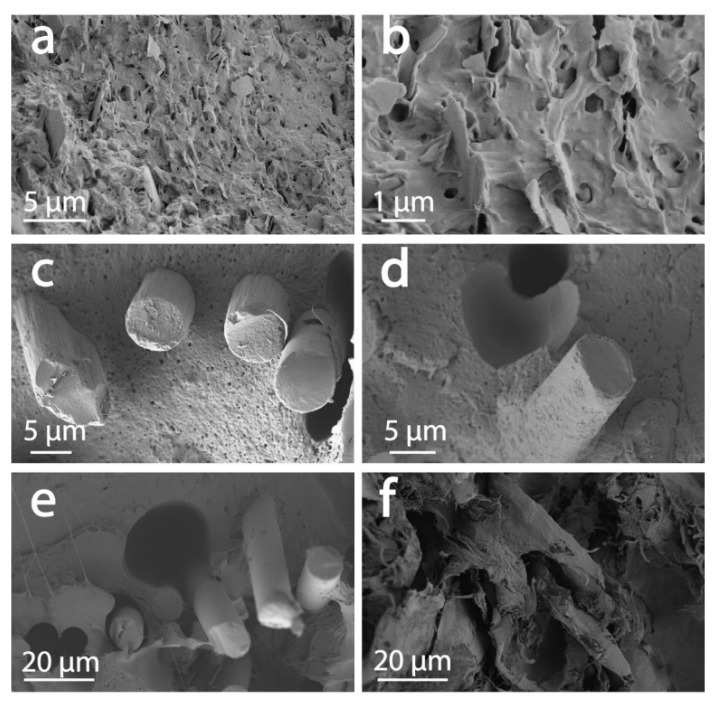
Magnified fractured surfaces as obtained after tensile tests highlighting polymer-additive interfaces observed by SEM for (**a,b**) Udiam-PLA, (**c,d**) CF-PET, (**e**) GF-PA6 and (**f**) UPM Formi-PLA.

**Figure 10 materials-13-00422-f010:**
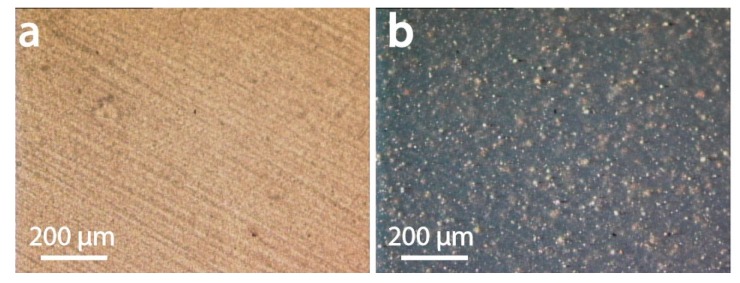
Transmission microscopy images obtained for (**a**) uDiam-PLA composites and (**b**) uDiam-PLA composites images with the samples between cross polarizers highlighting crystalline phase in uDiam-PLA (bright areas).

**Table 1 materials-13-00422-t001:** Descriptions, abbreviations, printing conditions and characteristics of the investigated printing filaments.

Material Description	Abbreviation in the Text	Brand	Nozzle Temperature (°C)	Bed Temperature (°C)	Suggested Print Speed (mm/s)	Density (g/cm^3^)
Polylactic acid	PLA	Ultimaker	210–230	60–75	40–80	1.24
Polyamide (grade PA6/66)	PA6/66	Ultimaker	230–260	60–70	40–80	1.14
Poly(ethylene terephthalate)	PET	Innofil3D	210–230	75	40–80	1.34
Cellulose polylactic acid	UPM Formi-PLA	Octofiber	210–225	60	15–30	1.21
Glass fiber polyamide (grade PA6)	GF-PA6	Owens-Corning	220–280	80–110	30–110	1.17
Carbon fiber poly(ethylene terephthalate)	CF-PET	Innofil3D	250–260	75	40–60	1.23
Nanodiamonds polylactic acid	Udiam-PLA	Carbodeon	220–250	Ambient	50–500	1.35

**Table 2 materials-13-00422-t002:** Thermogravimetric analysis of different printing filaments.

Sample	*WL*_220__°C_ (%)	*IDT* (°C)	*T_d_*_1_ (°C)	*WL*_1_ (%)	*T_d_*_2_ (°C)	*WL*_2_ (%)	*T_d_*_3_ (°C)	*WL*_3_ (%)	Ash (%)	Filler (%)
PLA	0.15	321.11	346.13	95.17	467.40	3.65	-	-	1.27	-
UPM Formi-PLA	1.10	310.61	343.28	85.20	443.05	13.50	-	-	1.20	14.70 ^a^
Udiam-PLA	0.87	334.85	356.61	81.93	460.48	3.01	-	-	14.62	14.62 ^b^
PA6/66	1.62	390.34	440.12	83.15	520.83	16.56	-	-	0.36	-
GF-PA6	1.07	393.34	433.75	62.25	489.70	8.29	-	-	29.50	29.50 ^c^
PET	0.19	381.47	420.56	84.24	526.80	15.59	-	-	0.08	-
CF-PET	0.15	386.08	423.26	69.97	508.63	13.18	698.18	15.358	0.39	15.74 ^d^

a = *WL*_2_ + Ash; b = Ash; c = Ash; d = *WL*_3_ + Ash.

**Table 3 materials-13-00422-t003:** Non-isothermal crystallization parameters of different printing filaments.

Sample	*T_g_*_(i)_ (°C)	*T_g_*_(ii)_ (°C)	*T_cc_*_1_ (°C)	Δ*H_cc_*_1_ (J/g)	*T_m_*_1_; *T_m_*_2_ (°C)	Δ*H_m_*_1+2_ (J/g)	*X_c_*_1*F*_ (%)	*X_c_*_13*D*_ (%)	*X_c_*_2*m*_ (%)	*T_onset_* (°C)	*T_mc_*_1_; *T_mc_*_2_ (°C)	*t*_1/2_ (min)	*R* (min^−1^)
PLA	58.76	58.22	113.99	18.978	147.78	21.279	2.46	22.71	21.30	-	-	-	-
UPM Formi-PLA	57.62	56.59	91.16	11.689	143.87; 151.57	23.257	14.47	29.10	19.71	40.04	31.38	0.87	1.15
Udiam-PLA	56.68	57.44	79.83	14.288	172.5	32.584	26.19	51.67	52.05	123.32	146.90; 116.7	0.66 *	1.50
PA6/66	46.39	56.78	-	-	180.59; 188.99	89.101	37.13	37.13	18.12	143.84	133.75	1.01	0.99
GF-PA6	34.06	67.11	-	-	207.61	47.481	29.28	29.28	24.80	176.19	170.8	0.54	1.86
PET	65.72	69.13	-	-	-	-	-	-	-	-	-	-	-
CF-PET	71.79	74.84	118.64	21.772	251.57	32.679	9.96	29.83	32.18	216.38	210.55	0.58	1.72

* *T_mc_*_2_ was used to calculate *t*_1/2_.

**Table 4 materials-13-00422-t004:** Mechanical and thermal characterization of investigated materials under tensile loading. Here, *E*, UTS, *ε*_UTS_, *ε*_Break_ and Δ*T*_Break_ refer to Young’s modulus, ultimate tensile strength, strain at ultimate tensile strength, strain at break and difference between the temperature at break and initial configuration, respectively. StDev. refers to the standard deviation.

Material	Cross-Section	*E*	UTS	*ε* _UTS_	*ε* _Break_	Δ*T*_Break_
	mm^2^	MPa	MPa	(mm/mm)	(mm/mm)	(°C)
PLA						
(Tests) Mean	42.39	1790.13	56.77	0.041	0.050	12.86
(Tests) StDev.	0.44	16.38	3.65	0.003	0.011	4.11
(Data sheet) [15]	-	2852.0	38.1	0.021	0.028	-
PET						
(Tests) Mean	42.38	1607.76	65.64	0.054	0.067	27.03
(Tests) StDev.	0.57	52.75	1.03	0.003	0.001	9.54
(Data sheet) [20]	-	2264.0	40.9	0.030	0.031	-
PA6/66						
(Tests) Mean	41.25	1299.77	58.29	0.061	-	-
(Tests) StDev.	0.31	9.44	0.20	0.001	-	-
(Data sheet) [15]	-	579.0	27.8	0.020	2.10	-
UPM Formi-PLA						
(Tests) Mean	41.89	809.91	16.69	0.033	0.038	5.56
(Tests) StDev.	0.15	30.73	0.38	0.004	0.004	1.83
(Data sheet) [17]	-	2600.0	28.0	0.050	-	-
GF-PA6						
(Tests) Mean	40.52	2533.22	91.16	0.051	0.051	14.41
(Tests) StDev.	0.13	19.88	0.78	0.002	0.002	2.86
(Data sheet) [18]	-	7400.0	102.0	0.021	0.021	-
CF-PET						
(Tests) Mean	41.57	1875.49	52.02	0.037	0.038	7.73
(Tests) StDev.	0.80	36.34	1.38	0.020	0.020	0.58
(Data sheet) [20]	-	9000.0	80.0	0.025	-	-
uDiam-PLA						
(Tests) Mean	41.56	1995.47	35.59	0.026	0.027	14.35
(Tests) StDev.	0.49	179.80	2.14	0.002	0.002	4.29
(Data sheet) [19]	-	6300.0	43.5	-	0.032	-

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
