# Peer review of "Comparative Screening of the Structural and Thermomechanical Properties of FDM Filaments Comprising Thermoplastics Loaded with Cellulose, Carbon and Glass Fibers"

_materials, 2020, doi:10.3390/ma13020422_

Round 1

Reviewer 1 Report

The investigation was carried out to examine the thermos-mechanical properties commercial polymer materials used in composite filaments. The obtained results are adopted to reveal new insights into the size morphology, and distribution of the constituents and inter phases of polymer filaments.

The SEM images are explained in detail to reveal the information about composites and their material behavior under strain. These evidence are useful for the readers to understand the fracture mechanisms (after tensile test).

But somehow, some figures has to be modified  for better understanding and looks better for readers.  

Please modify Figure 5 by removing inner grids and enlarge the image and also increase the font sizes in both axes. Due to compressed image, it is hard to read the data presented there.

Also please modify or rearrange some contents based on reader point of view. In some places it looks so confusing to understand what you are trying to say.

Paper can be considered for the publication after the modifications. 

Thank You.

Have a nice day.  

Reviewer 2 Report

Given the importance obtained by additive manufacturing recently, it is important to compare in a sound and professional way the performance of different filament materials over the range of temperature involved in their processing. This is wonderfully done in this paper. I can only suggest some minor modifications, which are reported later on.

The Abstract is very long with repetitions ("we investigate" twice, e.g.). In addition, the initial paragraph on the outstanding importance of FDM would be more appropriate in the Introduction with a number of references.

Line 104: "Layer height was taken to be 0.1 mm". Which was the accuracy of this assumption?

Line 410: please change "was" into "were"

Line 425: "wetted paper". Please give more details on which paper you used for that (areal weight, e.g., and was it absorbent?).

Figure 5: I would eliminate the decimal figure "0" also to enlarge the sub-figures and make them more visible. 

Figure 6: please provide the spatial dimensions of IR thermography images e.g., by a dimension bar   

Line 515: "The observed glass fibers are very narrowly dispersed in terms of diameter, ca. 5 μm". Do you mean they have on an average 5 micron diameter, or... please rephrase to make the statement more understandable. 

Line 518: please replace "thermos" with "thermo"

Please also check the names of materials for consistency throughout the paper: UPM Formi, instead of UPMFormi, e.g.

Reviewer 3 Report

The work is very interesting and very well written. It allows to have an overview about some different materials applicable in FDM process describing the main characteristics and behaviours.

I have few comments and suggestions: 

in section 2.3 the authors state that the tensile tests were carried out at room temperature. Table 4 reports the data about the characterization under tensile loading and a ΔTbreak is reported. I suggest to clarify in the manuscript what the ΔTbreak represents, since in this way can generate ambiguity. Figure 4. I kindly suggest to apply the same x scale in all the charts.  Figure 5. Like for Figure 4, I suggest to use the same scale in all the charts (both for x and y axis). I know that in same case can be "impossible", for example for y-axis in Fig. 5(c), but I guess that for sake of clarity in the other charts you can applied the same scale. Figure 6. Is it possible to report the same scale for the temperature? If not, I suggest to increase the size of the minimum and maximum values of the scale to have a faster comprehension.  Regarding the image about the PA6/66 in Figure 6. How can you explain the totally different profile  of the sample? Is this  phenomenon only related to the high dissipation of energy, or there are some other reasons?
